# Can *Miscanthus* Fulfill Its Expectations as an Energy Biomass Source in the Current Conditions of the Czech Republic?—Potentials and Barriers

Jan Weger [1,*], Jaroslav Knápek [2], Jaroslav Bubeník [1], Kamila Vávrová [1] and Zdeněk Strašil [3]

[1] Silva Tarouca Research Institute for Landscape and Ornamental Gardening, Public Research Institute (VÚKOZ), 252 43 Průhonice, Czech Republic; bubenik@vukoz.cz (J.B.); vavrova@vukoz.cz (K.V.)

[2] Faculty of Electrical Engineering (ČVUL FEL), Department of Economics, Management and Humanities, Czech Technical University in Prague, 166 27 Prague, Czech Republic; knapek@fel.cvut.cz

[3] Crop Research Institute, Public Research Institute (VÚRV), 161 00 Prague-Ruzyně, Czech Republic; pavelstrasil@mpsprojekt.cz

* Correspondence: weger@vukoz.cz; Tel.: +420-605-205-995

**Abstract:** Our article analyzes the main biological potentials and economic barriers of using *Miscanthus* as a new energy crop in agricultural practice in the Czech Republic and the Central-Eastern European region. We have used primary data from long-term field experiments and commercial plantations to create production and economic models that also include an analysis of competitive ability with conventional crops. Our results showed that current economic conditions favor annual crops over *Miscanthus* (for energy biomass) and that this new crop shows very good adaptation to the effects of climate change. Selected clones of *Miscanthus* × *giganteus* reached high biomass yields between 15–17 t DM ha$^{-1}$ y$^{-1}$ despite very dry and warm periods and low-input agrotechnology, and they have good potential to become important biomass crops for future bioenergy and the bioeconomy. Key barriers and factors are identified, including gene pool and agronomy improvement, change of subsidy policy (Common agriculture policy-CAP), climate change trends, and further development of the bioeconomy.

**Keywords:** *Miscanthus*; energy biomass; yields; invasive behavior; economics





## 1. Introduction

From the enlargement of the European Union (EU) in 2004 and 2007 to include ten Central and Eastern European (CEE) countries, the share of renewable energy sources in final gross consumption in the EU27 countries has almost doubled from 113 to 220 Mtoe, reaching 18.9% in 2018. Biomass continues to be the most important type of renewable energy source (RES); it contributed 59.2% to the total share of RES in final gross energy consumption in 2018 [1,2]. The total biomass utilization (for solid, liquid, and gaseous biofuels) has been growing in absolute terms since the EU enlargement (from 69 to 135 Mtoe) and is expected to continue to play a significant role in the gradual replacement of fossil fuels [3,4]. For instance, the supply of biomass from agricultural and perennial energy crops would need to increase by 29% to fulfill the ambitious goals of National Renewable Energy Action Plans [2]. These trends are expected to continue under The European Green Deal proposed by the current European Commission [5].

Of all the biomass types, solid biomass is currently most frequently used; in 2017, for the EU as a whole, 69% of the total contribution of biomass for energy from RES was solid biomass. One example of biomass being expected to contribute significantly to meeting 2030 targets is the Czech Republic, whose National Energy Action Plan foresees an increase in the share of RES in final gross consumption from about 15% (2019) to 22% in 2030 and the use of solid biomass for heating to increase by 26% (or 27 PJ) [3].

In CEE countries, of all the renewables, the one with the highest potential is biomass, especially when both residual and intentionally grown biomass sources on agricultural land have been included in the calculations [6]. Second-generation energy crops are an especially promising source for the future. These include selected clones or varieties of fast-growing trees (poplar, willow), perennial and some annual grasses (reed canary grass, *Miscanthus* ssp., triticale), and perennial rhizomatous plants (Schavnat—a spinach-sorrel hybrid, *Sida*—Virginia mallow). These non-food and high yielding crops have a much better energy input/output ratio (five to seven times) than first-generation crops like rape or cereals in regards to how much energy biomass is produced per hectare [7–9].

## 1.1. Experience with Miscanthus and Energy Crops

In previous analyses, it was expected that second-generation energy crops would be planted on larger areas of agricultural land [10–12] because of the growing demand for energy biomass, and the available political and financial support, although mostly indirect, e.g., subsidized price of electricity from intentionally grown biomass. It was expected that sufficient agricultural land for these plantations, including less productive lands for food crops [13], but also some better quality soils would be available because trends have shown that there is an overproduction of food crops (especially cereals) in the Czech Republic and other CEE countries [14,15].

Nevertheless, the potential of second-generation crops has not been realized as these crops are currently grown only on 0.14% of agricultural land in the Czech Republic. A similar situation can be found in other neighboring countries (Poland, Germany, Austria, Slovakia). For instance, in Germany, the land area suitable for *Miscanthus* has been assessed to be about 4 million ha [16], but the growing area is about 4000 ha [17], e.g., 0.03% of agricultural land. In Poland, according to the Polskie Towarzystwo Biomasy (POLBIOM, Warsaw) association a member of Bioenergy Europe (Brussels, Belgium), *Miscanthus* was grown on approximately 2000 ha from 2009–2011, but this number decreased significantly in the following years and was 733 ha in 2013 [18]. The current area could be even smaller (below 500 ha) because many farmers decided to cease *Miscanthus* plantations for energy biomass due to economic reasons [19]. From the experience of several European regions, it seems that the less favorable economic profitability of growing perennial *Miscanthus* for energy (direct burning) has been the crucial reason why many farmers have ceased planting it and returned to growing annual food crops.

Since the 1990s, many bioenergy projects have focused on planting second-generation crops for energy biomass (direct combustion or pyrolysis)—first, in Western and later, Eastern European regions [20–24]. In the Czech Republic, planting first started in the early 2000s with energy crops that can be easily planted, respectively seeded with standard agricultural mechanization, e.g., Schavnat and triticale (Figure 1) supported by relatively high national subsidies for the establishment of non-woody energy crops. By 2007, there were about 1200 hectares of these crops, but after full adoption of the EU's financial support scheme of the Common Agricultural Policy, when national subsidies no longer applied, the area of planted energy crops decreased. There were also often high losses in newly established plantations partially caused by the low quality of planting material and poor agrotechnology, unsuitable site selection, and sometimes, by extreme climate conditions.

Development of fast-growing trees and *Miscanthus* started in CEE countries after they entered the EU when legislative and 'agro-logistical' conditions (planting material, mechanization) have improved for farmers interested in new biomass crops. While dynamic planting of fast-growing trees (mainly poplar clone Max-4) continued until 2015 when new legislation on soil protection and decreased biomass co-firing has stopped this development, *Miscanthus* never became a 'new biomass crop' attractive enough for pioneering farmers (see Figure 1). The only large *Miscanthus* project in the Czech Republic was carried out by a joint-venture of *Miscanthus* Ltd., of the Pilsen energy company (Pilsen, Czech Republic) and regional forest enterprise, which was terminated after 3–4 years for various reasons, including failures in establishing *Miscanthus* plantations and slower growth than expected.

The Czech Republic's current planting area with *Miscanthus* is around 50 ha and is used mostly to produce animal bedding material, including pellets [25].

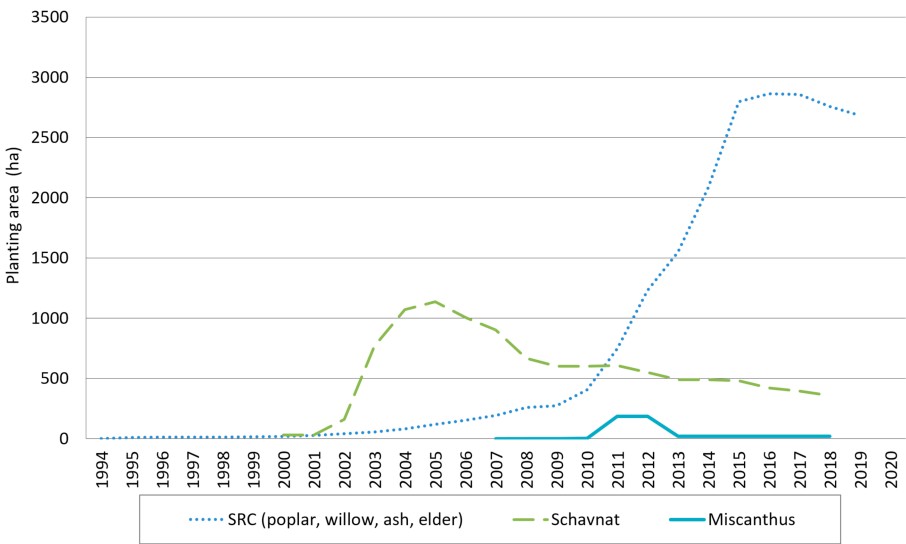

**Figure 1.** Development of planting area of selected second-generation energy crops in the Czech Republic (different sources compiled by Weger: primary data from annual reports of crops cultivated on soil blocks [MS Excel] of The State Agricultural Intervention Fund, Prague, Czech Republic.

### 1.2. Miscanthus Resources and Services

The species and varieties of genus *Miscanthus* Andersson consist of 15 taxa of C4 rhizomatous grasses [26]. It naturally occurs in tropical and moderate climate regions of eastern Asia and southeastern Africa. *Miscanthus* ssp. has been planted in many different world regions for ornamental, material, and energy purposes. Commercial and research organizations have produced many genotypes for these purposes. Present-day clones and varieties yield over 55 tons of raw biomass per hectare per year in favorable conditions [13,27]. Currently, the most commonly planted clone in Europe is *M. × giganteus* Greef et. Deu ex Hodkinson et Renvoize. It is a sterile triploid hybrid between diploid *M. sinensis* and tetraploid *M. sacchariflorus* [28,29].

*Miscanthus* straw contains about 41–45% cellulose, 20.6–33% hemicellulose and 19–23.4% lignin [30,31] and has a good heating value (17.6 GJ $t^{-1}$ at 0% moisture respectively 13.60 GJ $t^{-1}$ at average harvesting moisture of 20%). It is also a suitable source for the production of pulp [32]. *Miscanthus* biomass can also be used to produce construction materials [21,33,34] or composted together with cattle or pig slurry [35].

Due to *Miscanthus'* environmental benefits (soil protection, crop diversification), it is especially a suitable alternative in places where food crops are not productive or planted. *Miscanthus*, similar to fast-growing trees [36], does not require high doses of industrial fertilizers [37–39], nor does it tend to be susceptible to diseases or harmful organisms [40,41]. Lower levels of applied fertilizers and pesticides decrease the risk of soil and groundwater contamination. The risk of soil erosion from water or wind in *Miscanthus* plantations is serious only during the first year or two after establishment when the root system has not fully developed, and production of phytomass, especially leaf litter, covering the soil surface is limited. *Miscanthus* can also be planted on grasslands with no-tillage agronomy, thus preventing the loss of biodiversity and soil carbon during establishment [42].

There are also ecological reasons for growing *Miscanthus*, e.g., decreasing the difference between the lowest and highest soil temperatures; facilitating better soil-water management; reducing soil erosion from water and wind; possibly improving the content of the soil's organic matter [43–48].

The main risks for the successful establishment of plantations include low quality of rhizomes, insufficient weeding, site selection (dry or waterlogged) and extreme winter temperatures below minus 3.5 °C in the rhizome zone of the soil [49].

### 1.3. Invasive Risks of Miscanthus

*Miscanthus sinensis* has been registered as a weedy or invasive plant in many regions worldwide, e.g., the USA, Canada and Australia [50,51]. An example of its invasiveness can be found in the results of [52], who state that in all experimental field plots, nearly all species and cultivars in the Trinity College Botanic Garden collection in Dublin (Ireland) create viable seeds. Seed viability was also confirmed in genotypes that were considered to be sterile. Seed germination tests of selected decorative *Miscanthus* clones proved that not all genotypes of *Miscanthus sinensis* [29] and other species have the same invasive potential [53]. For example, in eastern parts of the USA, the relatively high invasive potential of self-established local populations of *Miscanthus sinensis* was thoroughly studied; at least four of six populations have origins in nearby growths of ornamental *Miscanthus* plants that were established 20 to 120 years before [54]. From the invasiveness perspective, *Miscanthus × giganteus*, a sterile triploid hybrid, is not considered a risky crop [55,56]. Only a few publications mention the uncontrolled spreading or escape of *Miscanthus* ssp. plants, especially *M. sinensis* and *M. sacchariflorus* from field growths in the Czech Republic and Central Europe [57–60], but more articles can be found in other mild-climate countries.

### 1.4. Objectives of the Article—Potentials and Barriers of Miscanthus as an Energy Crop

Despite the positive production and environmental characteristics partially described above, several barriers and limitations have prevented *Miscanthus* from becoming suitable as a (still) new crop for energy biomass in the Czech and European conditions.

Therefore, in the article, we have focused on the key aspects of *Miscanthus* plantations—production, invasive risk, and economy—to identify the main potentials, barriers, and recommendations for *Miscanthus* to develop as a standard agriculture crop in future farming practice for energy or possibly other sectors of the bioeconomy.

The countries of the CEE region have similar economic conditions—especially the cost of human labor, land costs, and at the same time, have similar conditions in terms of agricultural subsidies. If we assume similar climatic conditions and the use of similar agrotechnical procedures, then similar conclusions can be expected in terms of the economic efficiency of *Miscanthus* cultivation.

## 2. Materials and Methods

Different methodologies, as described in the following chapters, were used to evaluate the main potentials and risks of producing *Miscanthus*. We have mainly analyzed primary data from our field experiments to evaluate production potential and invasive risk. These data were then compared and confirmed with data from commercial plantations and scientific literature from areas with similar growing conditions.

To analyze the economic efficiency of *Miscanthus* cultivation, we use a modeling approach based on the real economic conditions in the Czech Republic at the price level of 2019. We have also added the aspect of competition with conventional agricultural crops into our economic model.

Since 1989, numerous field experiments have been established to evaluate yield and growth of *Miscanthus* species and genotypes in EU countries within European international programs and projects (JOULE, FAIR) [49]. The European *Miscanthus* Improvement Project (EMI, 1996–1999) also started a breeding program to improve yield and growth of this 'novel crop' for European conditions. Based on the EMI results and with support of the consortium members, we have established experiments to evaluate *Miscanthus* potential in Czech environmental and economic conditions—first, the genotype collection at the Průhonice-Zelinářská zahrada location in 2002 and five years later, the clonal test at two

locations: Průhonice-Michovky and Lukavec in 2007 (Figure 2). The distance between the genotype collection and the clonal field experiment in Průhonice is 1.4 km.

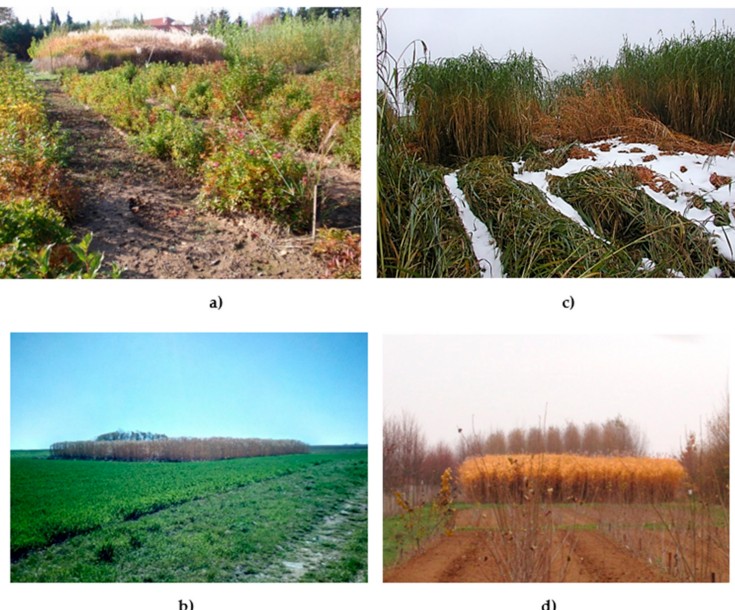

**Figure 2.** Pictures of experimental sites with *Miscanthus*: (**a**) Genotype collection Průhonice-Zelinářská zahrada with marked seedlings of *Miscanthus sinensis* invading the neighboring nursery in 2008; (**b**) Clonal experiment in Lukavec before spring harvest 2019; (**c**) Clonal experiment in Lukavec—lodging of *Miscanthus sinensis* (clone M4 GOFAL) under early snow in October 2009; (**d**) Clonal experiment in Průhonice-Michovky in October 2014.

## 2.1. Climatic and Soil Conditions of Experimental Sites

Soil and climatic conditions at both locations, Lukavec and Průhonice, are described in Table 1. Experiments in Průhonice are located between 310–330 m above sea level at a flat area and without exposition. Soil is Cambisol [61]. The mean year temperature is 8.8 °C. Mean sum of precipitation is 580 mm. The Průhonice sites are in a region where cereals are produced, while Lukavec is a potato-producing region. Changes in weather parameters during the experiment and their comparison using long-term temperature and precipitation averages (1961–1990) [62] are shown in Table 2.

**Table 1.** Site conditions at locations of experiments (before planting, 2007, respectively 2002).

| Factor | Průhonice Michovky | Průhonice Zelinářská Zahrada | Lukavec |
|---|---|---|---|
| Latitude | 49°59′ | 49°59′ | 49°34′ |
| Longitude | 14°34′ | 14°34′ | 14°59′ |
| Altitude (m) | 332 | 310 | 570 |
| Soil texture | clay-loess | clay-loess | sandy-loam |
| Soil type | Cambisol | Cambisol | Cambisol |
| Mean year temperature (°C) | 8.8 | 8.8 | 7.3 |
| Mean year sum of precipitation (mm) | 580 | 580 | 682 |
| **Agrochemical properties of soil (before establishment):** | | | |
| Content of humus (%) | 1.0 | 1.3 | 3.4 |
| pH ($H_2O$) | 6.22 | 7.11 | 6.14 |
| Content of P (Mehlich III, mg.kg$^{-1}$) | 54 | 395 | 112 |
| Content of Mg (Mehlich III, mg.kg$^{-1}$) | 179 | 190 | 114 |
| Content of K (Mehlich III, mg.kg$^{-1}$) | 143 | 354 | 337 |

**Table 2.** Annual temperatures and precipitations in Průhonice and Lukavec between 2007–2017.

| No. | Year | Average Annual Temperature | | Annual Sum of Precipitation | |
|-----|------|------|------|------|------|
| - | | Průhonice | Lukavec | Průhonice | Lukavec |
| | | (°C) | (°C) | (mm) | (mm) |
| 1 | 2007 | 10.2 | 8.5 | 517 | 777 |
| 2 | 2008 | 9.8 | 8.6 | 502 | 604 |
| 3 | 2009 | 9.4 | 8.4 | 599 | 788 |
| 4 | 2010 | 8.0 | 7.2 | 764 | 940 |
| 5 | 2011 | 9.6 | 8.4 | 563 | 670 |
| 6 | 2012 | 9.5 | 8.1 | 553 | 724 |
| 7 | 2013 | 9.0 | 7.3 | 667 | 876 |
| 8 | 2014 | 10.6 | 8.6 | 548 | 707 |
| 9 | 2015 | 10.7 | 8.7 | 427 | 576 |
| 10 | 2016 | 10.0 | 7.9 | 499 | 601 |
| 11 | 2017 | 10.0 | 7.9 | 553 | 777 |
| 12 | 2018 | 11.0 | 8.8 | 354 | 509 |
| 13 | 2019 | 10.7 | 9.1 | 521 | 680 |
| | Average | 9.9 | 8.3 | 544 | 710 |

### 2.2. Assortment and Design of the Clonal Experiment

The clonal experiment was established with four clones of *Miscanthus sinensis* and two clones of the triploid hybrid *Miscanthus × giganteus*. The clones come from the national collection (Table 3). The experiment was planted using *Miscanthus* rhizomes that were at least 70 mm long. As shown in Figure 3, the experiment's design includes 4 random repetitions, a 1 × 1 m planting scheme and a density of one plant per 1 m². There are 18 plants of one clone in an individual plot, respectively 36 plants for clones M1 and M6 (double-sized plots). Additionally, the clone *Miscanthus × giganteus* (M12) from the Crop Research Institute in Praha-Ruzyně was planted as an isolation row around the experiment. Plants were measured every year: number and height of plants, number of stems, and fresh weight of biomass (straw) of each plant. Twice as many rhizomes of clones M1 and M6 were planted (36 plants per plot) to compare autumn and spring harvest.

**Table 3.** Assortment of *Miscanthus* clones included in the experiments.

| Clone No. | Clone Code | Taxonomical Classification | Origin | Number of Individuals | Number of Individuals |
|-----------|-----------|------|------|------|------|
| - | Clonal field experiment | | | Průhonice Michovky | Lukavec |
| M1 | M-GigM53-003 | *M. × giganteus* | Germany | 144 | 144 |
| M2 | M-GigFou-009 | *M. × giganteus* | Denmark | 72 | 72 |
| M3 | M-sin902-005 | *M. sinensis* | Denmark | 72 | 72 |
| M4 | M-sinGOF-002 | *M. sinensis* | Germany | 72 | 72 |
| M5 | M-sin903-006 | *M. sinensis* | Denmark | 72 | 72 |
| M6 | M-sinM43-004 | *M. sinensis* | Germany | 144 | 144 |
| M12 ** | M-GigVUR-012 | *M. × giganteus* | Czech Rep. | 100 | 100 |
| - | Genotype collection | | | Průhonice Zelinářská zahrada | - |
| M7 * | M-sin101-007 | *M. sinensis* | Denmark | 27 | - |
| M8 * | M-sin906-008 | *M. sinensis* | Denmark | 27 | - |
| M9 * | M-GigFou-009 | *M. × giganteus* | Denmark | 27 | - |
| M10 * | M-sacHon-010 | *M. sacchariflorus* | Denmark | 27 | - |
| M11 * | 'Goliath' | *M. sinensis* | Czech Rep. | 27 | - |
| M13 * | M-sinJes-001 | *M. sinensis* | Germany | 27 | - |

* Additional clones in the genotype collection used for monitoring invasive behavior; ** used only in isolation row of clonal field experiment (Průhonice-Michovky).

**Figure 3.** Design of field experiments used for data collection: (**a**) Scheme of the clonal experiment in Lukavec; (**b**) Scheme of the clonal experiment in Průhonice-Michovky; (**c**) Genotype collection Průhonice-Zelinářská zahrada; rept.1–4 are repetitions of field experiments.

### 2.3. Establishment and Maintenance of the Clonal Experiment

The experimental field was plowed and harrowed in autumn 2006 and leveled in spring 2007 to be ready for manual planting of *Miscanthus* rhizomes in May 2007. The design of the experiment was in a semi-randomized block design (Figure 3). Rhizomes were collected from the genotype collection a few days before planting and planted on the 4th May in Lukavec and 15th May 2007 in Průhonice.

For *Miscanthus* × *giganteus*, we used one standard rhizome (70–80 mm) and for *Miscanthus sinensis*, which has thinner rhizomes, we used two rhizomes in one 3–5 cm deep hole. In the first and second years after planting rhizomes, the plantations were weeded manually. Herbicides were not used. Later, when the *Miscanthus* plants grew fast enough, they did not need any weeding. Potential weeds were also suppressed by rich leaf fall.

As we aimed to study *Miscanthus* production with minimized inputs, no fertilizers were used at the Průhonice sites before establishment and during growth at both sites. The Lukavec site field was fertilized once before establishment using 70 kg ha$^{-1}$ K (potassium salt) and 40 kg ha$^{-1}$ P (superphosphate).

### 2.4. Evaluation of Yield and Biometric Characteristics

The following biometric parameters were measured in the clonal experiment: number of stems, the height of plants, and volumes of fresh biomass. The number of plants was counted before harvests (autumn and spring). The height of individual plants was measured from the ground surface to its highest point (of straw) in an upright position. All biometric parameters were measured a few days before the autumn and spring harvests.

Harvests were performed at both sites, usually between November–December for autumn harvest and March–April for spring harvest. In autumn and spring, individual plants were first cut down using a brush cutter, and immediately, each plant was

weighed on digital scales with an accuracy of ±5 g to obtain fresh weight. One fresh biomass sample was taken from each harvested clone to analyze moisture content and calculate dry mass yield. These samples were dried at 105 °C until constant weight. At this temperature, the energy (fuel) characteristics of biomass are not influenced. In our conditions, drying usually lasted 1.5–2 days. Yields of dry biomass were calculated from the field weight of fresh biomass multiplied by dry matter content in samples from each clone. Harvested biomass was calculated in tons of dry biomass per hectare and year [63,64].

*2.5. Modeling the Economic Efficiency of Miscanthus Plantations*

The life span of *Miscanthus* plantations is expected to range from 8–14 years. *Miscanthus* plantations characteristically have high costs of establishment and do not reach maximum yields immediately after establishment, but only after several years. Thus, simple calculation methods based on annual yields and costs, such as for conventional crops, cannot be used to assess the economic efficiency of *Miscanthus* cultivation [65]. The evaluation must include the whole, relatively long, life cycle of this energy crop—from land preparation, stand establishment, crop maintenance throughout the life of the stand (e.g., fertilization), to harvest of biomass and finally, stand eradication at the end of the stand's life followed by site restoration.

Using a methodology based on simulating net cash flows throughout the planted energy crop's life cycle is appropriate for assessing the economic efficiency of *Miscanthus* cultivation [66]. This standard procedure is based on the calculation of the net present value (NPV) of the project [67], not including the NPV directly, but rather, counting the so-called minimum price of production (here, the minimum price of biomass produced), which will provide the investor with the required return on his investment. Therefore, the minimum price calculation is based on finding a biomass price such that NPV = 0 (see equation (1)). The investor then produces a return on the capital transferred at the discount rate.

$$c_{min} \; : \; NPV = \sum_{t=1}^{T_h} CF_t \cdot (1 + r_n)^{-t} = \sum_{t=1}^{T_h} (c_{\min,t} \cdot Q_t + S_t - E_t) \cdot (1 + r_n)^{-t} = 0 \qquad (1)$$

where

$c_{min,t}$—minimum price of biomass in year t (EUR/GJ)
$Q_t$—biomass production measured in heat energy (GJ)
$S_t$—project subsidies in year t (EUR)
$E_t$—project expenditure in year t (EUR)
$r_n$—nominal discount ($-$)
$T_n$—evaluation period (here, 10 years)
$CF_t$—cash flow in year t (EUR).

Note: For practical reasons, the minimum price of biomass is expressed in monetary units per GJ (at moisture levels at harvest). This then also allows a direct comparison of the minimum price of different energy crops, irrespective of their moisture levels.

For analysis purposes, we created a model of a 10 ha *Miscanthus* plantation that reflects the typical conditions of growth of this energy crop in the Czech Republic. The model is based on the assumption of a rigorous valuation of all costs at market prices (2019 price level), an estimation of the scope of individual activities according to the data obtained in field trials on experimental sites, and respecting the time value of money. The model uses long-term average inflation of 2%, a nominal discount of 10%, and an income tax of 19%. All the financing of the project is assumed to be from the investor's own resources.

The model works with 6 different yield curves (Yc) reflecting expected yields at different site conditions (climatic and soil)—see Figure 4.

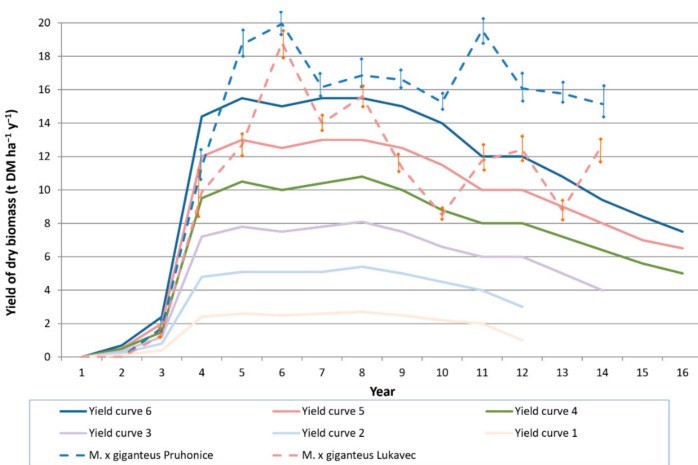

**Figure 4.** Dry biomass yields of *Miscanthus* × *giganteus* (average with standard error bars of clones M1, M2) from plantations in Průhonice-Michovky and Lukavec, and yield curves (Yc 1–6) of expected yields in commercial practice created for different growing zones for *Miscanthus* in the Czech Republic; growing years (1–16) are considered as financial years (e.g., from January to December).

For modeling purposes, the *Miscanthus* crop processes have been divided into the following:

1. Project and land preparation: Land preparation in the autumn before spring planting includes moderate deep plowing and harrowing, fertilization with NPK (eventually, lime) according to the land's condition. Pre-fertilization in the form of NPK is equivalent to approximately 60 kg N/ha.
2. Costs of establishing a stand: Planting 8000 rhizomes per hectare using a potato planter, post-emergence weeding using a herbicide (eradication of dicotyledonous weeds after one year of *Miscanthus* growth).
3. Planting material: Price used is 0.12 EUR/*Miscanthus* rhizome, which has been typical for a larger amount of purchased material.
4. Harvesting and processes between harvests: Harvesting (bales 80 × 90 cm) takes place in the winter season, with a *Miscanthus* moisture content of 20% at harvest and a calorific value of 13.75 GJ/t (raw biomass). Yield curves already respect the assumed losses of biomass due to the winter harvest. Fertilizer costs (60 kg N/ha in NPK) are estimated from experience with experimental plots once every three years. After the fifth harvest, Ca fertilizing with approximately 2–2.5 tons of dolomitic limestone per hectare is expected.
5. Crop management, subsidies, and land rent: Rent for land is assumed to be 200 EUR/ha/year (approximate median Czech cost of land rent), and overheads are estimated at 40 EUR/ha/year.
6. Costs of stand eradication: After the tenth harvest, the crop is eradicated by deep plowing.
7. Subsidy: Single Area Payment Scheme (SAPS), including a Greening Payment is approximately 210 EUR/ha/year.

## 3. Results

### 3.1. Climatic Conditions During Experiments

From 2007 to 2019, the mean annual temperatures in Průhonice ranged between 8.0 °C (2010) and 11.0 °C (2018), while in Lukavec, between 7.2 °C (2010) and 9.1 °C (2019). In Průhonice during the same period, annual sums of precipitation ranged from 354 mm (2018) to 764 mm (2010), while in Lukavec, it was 509 mm (2018) and 940 mm (2010). Compared with the climatological long-term normal (1960–1991) from the Czech Hydrometeorological Institute in Prague [68], temperatures in Průhonice were from normal to extraordinary above normal, while for Lukavec, from normal to above normal. Precipitations ranged

from very below normal to above normal in Průhonice, while in Lukavec, from normal to very above long-term normal. The year 2018 was the driest at both localities during the experiment (354 mm respectively 509 mm) and over the last 30 years. In comparison with the long-term normal, the weather in Průhonice was warmer (annual average daily temperatures +1.7 °C) and drier (deficit precipitations minus 661 mm), and the weather in Lukavec was also warmer (annual average daily temperatures +1.1 °C) and precipitation was higher than normal (surplus of precipitations 842 mm) within 13 years of the experiment.

### 3.2. Survival and Lodging

The survival rate of tested *Miscanthus* clones was quite good on both sites, reaching 90% in Lukavec respectively 89% in Průhonice-Michovky, with only the M5 clone exceeding 20% losses in Lukavec. Survival rates remained stable after the first two years—percentages of living plants are shown in Table 4. Some tested clones have shown insufficient adaptability to winter snow cover, causing serious damage by lodging the whole plant. In the case of clone M6, which has been preselected from the genotype collection as well growing, lodging caused by snow led to reduced yields and loss of biomass quality (Figure 2b).

**Table 4.** Survival rates (%) of tested *Miscanthus* clones in clonal experiments in Průhonice-Michovky and Lukavec in the year of establishment (2007) and after ten years (March 2017).

| Clone | Průhonice Michovky 2007 (XII) | Průhonice Michovky 2017 (III) * | Lukavec 2017 (III) |
|---|---|---|---|
| M1 | 84 | 88 | 85 |
| M2 | 88 | 99 | 85 |
| M3 | 88 | 100 | 96 |
| M4 | 93 | 88 | 79 |
| M5 | 85 | 99 | 99 |
| M6 | 95 | 99 | 94 |
| Average | 89 | 96 | 90 |

* After the first year, dead plants were replaced in the experimental plantation in Průhonice-Michovky.

### 3.3. Biomass Yields

The mean yield of dry biomass of all clones of *Miscanthus* after 13 (spring harvests) from both sites was 10.03 t dry matter (DM) ha$^{-1}$ y$^{-1}$, respectively 10.80 t DM ha$^{-1}$ y$^{-1}$ at Průhonice-Michovky and 9.27 t DM ha$^{-1}$ y$^{-1}$ at Lukavec. Yields from spring harvests of all clones in the experiment are shown in Table 5. Yields from the establishment year (2007, harvested in spring 2008) were not calculated because the development of *Miscanthus* plants was slow (1–3 stems below 1 m per plant), and biomass harvests are not performed in practice.

The best yielding clones were M2 with 14.7 t DM ha$^{-1}$ y$^{-1}$ and M1 with 13.5 t DM ha$^{-1}$ y$^{-1}$ after 12 spring harvests from the Průhonice-Michovky site. Both clones are *Miscanthus* × *giganteus*. At the Lukavec site, the same clones M1 and M2 reached yields of 10.8 t DM ha$^{-1}$ y$^{-1}$ (M1) and 10.4 t DM ha$^{-1}$ y$^{-1}$ (M2). The best yielding *Miscanthus sinensis* clones were M4 and M5. After the last harvest in spring 2020, M5 is the highest yielding *Miscanthus sinensis* clone at Lukavec (10.2 t DM ha$^{-1}$ y$^{-1}$), while M4 is the highest yielding clone at the Průhonice site (12.7 t DM ha$^{-1}$ y$^{-1}$).

At the Průhonice-Michovky site, yields of clones M1, M2, M4 and M5 continuously increased for four years. At the Lukavec site, harvests of clones M5 and M6 increased for three years, while clones M1, M2, M3, M4 increased for four years.

Our statistical examination provided the following results: At the Lukavec site, even though Miscanthus × *giganteus* had higher yields than *Miscanthus sinensis*, no statistical difference was found between yields of these species due to the high variability of data. At the Průhonice-Michovky site, clones of *Miscanthus* × *giganteus* (M1, M2) grew better than those of Miscanthus *sinensis* (M3, M5, M6, M4).

During the experiment's existence at the Lukavec site, the highest harvest yield of dry biomass was reached in autumn (19.1 t DM ha$^{-1}$ y$^{-1}$ in 2011) by clone M1. The highest yield of dry biomass at the Průhonice-Michovky site (21.1 t DM ha$^{-1}$ y$^{-1}$) was reached in autumn 2011, also by clone M1.

**Table 5.** Dry biomass yields (t DM ha$^{-1}$ y$^{-1}$) of *Miscanthus* clones in spring harvest in clonal experiment at Lukavec and Průhonice-Michovky (2008–2017).

| Lukavec | | | | | | | | | | | |
|---|---|---|---|---|---|---|---|---|---|---|---|
| **Year** [†] | **M1** | **s** | **M2** | **s** | **M3** | **s** | **M4** | **s** | **M5** | **s** | **M6** | **s** |
| 2008 * | - | - | - | - | - | - | - | - | - | - | - | - |
| 2009 | 1.2 AB | 0.45 | 1.4 ABC | 0.86 | 2.3 C | 0.94 | 1.1 A | 0.73 | 2.3 BC | 0.58 | 1.9 ABC | 0.62 |
| 2010 | 9.3 A | 3.63 | 10.4 | 4.69 | 7.7 | 3.18 | 7.5 | 5.03 | 10.5 | 0.54 | 10.0 A | 1.44 |
| 2011 | 11.2 A | 0.75 | 14.2 | 1.35 | 9.4 | 0.80 | 10.9 | 3.33 | 11.3 | 0.57 | 14.0 B | 1.62 |
| 2012 | 19.1 D | 1.74 | 18.5 D | 2.75 | 10.9 A | 1.45 | 16.2 CD | 3.49 | 14.3 BC | 1.06 | 12.4 AB | 0.54 |
| 2013 | 14.2 B | 0.91 | 13.8 | 1.73 | 10.2 | 1.51 | 12.8 | 2.88 | 17.1 | 6.72 | 11.0 A | 1.75 |
| 2014 | 16.3 E | 1.46 | 14.9 DE | 2.16 | 8.0 A | 1.75 | 12.4 CD | 2.71 | 11.5 BC | 1.84 | 9.4 AB | 1.37 |
| 2015 | 11.6 C | 1.32 | 11.0 C | 1.82 | 6.4 A | 1.34 | 9.6 BC | 2.10 | 10.3 C | 1.99 | 7.3 AB | 0.66 |
| 2016 | 8.5 A | 2.13 | 8.6 A | 1.57 | 7.9 A | 0.45 | 11.4 BC | 2.42 | 14.2 D | 1.93 | 10.4 B | 0.78 |
| 2017 | 12.4 D | 2.5 | 11.1 C | 1.9 | 6.2 A | 1.7 | 9.8 C | 1.0 | 8.8 BC | 4.3 | 7.7 B | 0.9 |
| 2018 | 13.8 B | 2.21 | 11.0 AB | 1.03 | 7.9 A | 3.06 | 10.8 AB | 3.62 | 13.3 B | 2.64 | 9.0 A | 0.94 |
| 2019 | 9.3 AB | 1.68 | 8.3 A | 1.66 | 8.0 A | 2.02 | 7.8 A | 1.54 | 11.8 B | 3.01 | 7.8 A | 1.42 |
| 2020 | 13.2 B | 2.75 | 12.2 B | 0.67 | 7.6 A | 1.23 | 13.1 B | 1.00 | 11.8 B | 1.60 | 8.2 A | 0.95 |
| Average ** | 10.8 | | 10.4 | | 6.9 | | 9.2 | | 10.2 | | 8.1 | |
| **Průhonice-Michovky** | | | | | | | | | | | | |
| **Year** | **M1** | **s** | **M2** | **s** | **M3** | **s** | **M4** | **s** | **M5** | **s** | **M6** | **s** |
| 2008 * | - | - | - | - | - | - | - | - | - | - | - | - |
| 2009 | 2.1 AB | 0.93 | 1.4 A | 0.86 | 1.9 AB | 1.14 | 1.3 A | 0.78 | 1.5 A | 0.59 | 3.5 B | 1.59 |
| 2010 | 12.2 B | 2.64 | 10.6 AB | 2.47 | 7.7 A | 1.76 | 9.2 AB | 3.09 | 8.0 A | 1.36 | 11.7 B | 1.54 |
| 2011 | 17.5 D | 2.10 | 19.9 D | 1.36 | 6.0 A | 0.95 | 14.1 C | 1.77 | 8.1 AB | 0.44 | 9.9 B | 2.33 |
| 2012 | 18.8 B | 1.48 | 21.1 B | 1.97 | 8.6 A | 0.87 | 19.3 B | 3.24 | 10.5 A | 1.04 | 10.8 A | 1.63 |
| 2013 | 15.4 B | 2.19 | 16.9 B | 1.60 | 9.1 A | 1.16 | 16.1 B | 2.40 | 9.9 A | 1.04 | 9.2 A | 1.41 |
| 2014 | 15.8 CD | 1.91 | 17.9 D | 2.22 | 8.6 AB | 0.86 | 14.0 C | 1.28 | 10.6 B | 1.02 | 8.0 A | 1.25 |
| 2015 | 15.9 CD | 2.00 | 17.3 D | 1.18 | 7.9 AB | 0.68 | 14.9 C | 1.36 | 10.0 B | 1.18 | 7.3 A | 2.18 |
| 2016 | 14.4 C | 0.97 | 16.1 C | 0.86 | 7.1 A | 1.52 | 14.1 C | 1.21 | 10.3 B | 1.26 | 10.0 B | 0.97 |
| 2017 | 18.4 CD | 2.1 | 20.7 D | 0.99 | 9.1 A | 0.8 | 15.5 C | 1.39 | 11.6 B | 1.36 | 9.3 A | 1.89 |
| 2018 | 14.5 CD | 1.89 | 17.6 DE | 1.60 | 7.5 A | 1.59 | 18 E | 3.25 | 12.4 BC | 2.56 | 9.7 AB | 1.12 |
| 2019 | 14.9 B | 1.94 | 16.7 B | 1.54 | 7.5 A | 1.61 | 14.4 B | 2.60 | 9.7 A | 2.06 | 8.5 A | 0.64 |
| 2020 | 15.3 B | 2.06 | 14.9 B | 3.46 | 9.8 A | 0.73 | 14.3 B | 1.36 | 10.0 A | 2.24 | 9.3 A | 1.34 |
| Average ** | 13.5 | | 14.7 | | 7.0 | | 12.7 | | 8.7 | | 8.3 | |

Table legend: s—standard deviation; * yield was not evaluated due to slow growth in the year of the experiment establishment (2007); [†] considered as financial years from January to December (e.g., yield from spring harvest in 2008 represents biomass from the 2007 vegetative period); [A–E] indexed capital letters show results of statistical analysis of yields of tested clones in individual years (MANOVA–Duncan; homogenous groups); ** average yields from 12 harvests are calculated for 13 years of the experiment.

The expected yields of *Miscanthus*, expressed in yield curves (Figure 4), are linked to the site's soil and climatic conditions. The expected yield can be understood as the long-term average yield at the given age of the crop plantation in areas with similar climatic-soil conditions while respecting proper cultivation conditions.

Yield curves and growing zones for *Miscanthus* as well as other energy crops in the Czech Republic were created by energy crop experts. These experts used field data (yields) from long-term experimental plots and commercial plantations, and the Czech typology of agricultural land for agriculture production [69] that contains climatic, soil and site conditions of each farm field in the Czech Republic, i.e., the Valuation soil ecological unit. By combining these characteristics with empirically measured yields, six land suitability types and growing zones have been deduced for *Miscanthus*. Yield curves for each growing zone were then created using yields from consecutive harvests from

long-term experimental plots and commercial plantations. A more detailed description of the methodology used to create yield curves and growing zones for *Miscanthus* can be found in [10,66,70–72].

### 3.4. Biomass Parameters in Autumn and Spring Harvests

Comparison of dry biomass yields (t DM ha$^{-1}$ y$^{-1}$) from the autumn and spring harvests for *Miscanthus* clones M1 (*Miscanthus* × *giganteus*) and M6 (*Miscanthus sinensis*) at Průhonice-Michovky and Lukavec in consecutive harvests have shown that spring harvest yields are between 18–31% lower than the autumn harvest yields, depending on the clone and site (see Table 5 for spring harvest and Table 6 for autumn harvest). Differences between autumn (November) and spring (April) yields in individual harvests of both clones have been diverse, predominantly because of the course of winter weather and the occurrence of biotic and abiotic damage (lodging by snow, animals—Figure 2b) in the individual years.

Moisture content in harvested biomass varied between 7–36% (Ø 15%) in the spring harvest and between 20–62% (Ø 43%) in the autumn harvest depending on clone (M1 and M6) and year (weather before harvest) (see Table 6).

**Table 6.** Dry biomass yields in the autumn harvest and moisture content in biomass from autumn and spring harvests of *Miscanthus* clones M1 and M6 in the clonal experiment (Lukavec, Průhonice-Michovky).

| Biomass yields in the Autumn Harvest (t (DM)ha$^{-1}$ year$^{-1}$) | | | | | | | |
|---|---|---|---|---|---|---|---|
| | **Lukavec** | | | | **Průhonice-Michovky** | | |
| **Year** [†] | **M1** | **s** | **M6** | **s** | **M1** | **s** | **M6** | **s** |
| 2008 | 3.0 [A] | 1.18 | 3.7 [A] | 0.73 | 3.1 [A] | 1.53 | 4.5 [A] | 1.74 |
| 2009 | 10.1 [A] | 0.87 | 12.3 [B] | 2.69 | 13.0 [A] | 2.74 | 13.2 [A] | 0.19 |
| 2010 | 19.1 [A] | 4.81 | 15.9 [A] | 1.19 | 19.6 [B] | 2.57 | 13.2 [A] | 2.76 |
| 2011 | 23.6 [B] | 1.34 | 14.8 [A] | 0.84 | 21.3 [B] | 3.68 | 13.7 [A] | 1.68 |
| 2012 | 21.6 [B] | 1.07 | 16.2 [A] | 1.18 | 19.2 [A] | 5.62 | 12.2 [A] | 2.21 |
| 2013 | 29.1 [B] | 1.26 | 17.8 [A] | 1.52 | 21.4 [B] | 4.29 | 12.4 [A] | 2.51 |
| 2014 | 21.5 [B] | 1.85 | 13.5 [A] | 1.86 | 18.2 [A] | 4.30 | 13.8 [A] | 1.28 |
| 2015 | 14.0 [A] | 1.20 | 12.6 [A] | 1.00 | 16.8 [A] | 3.45 | 12.7 [A] | 3.05 |
| 2016 | 13.0 [A] | 1.47 | 13.2 [A] | 2.37 | 22.4 [B] | 3.30 | 16.0 [A] | 2,52 |
| 2017 | 16.4 [B] | 1.23 | 12.4 [A] | 0.84 | 19.9 [B] | 3.93 | 11.9 [A] | 1,51 |
| 2018 | 12.8 [A] | 2.25 | 14.1 [A] | 1.30 | 17.0 [B] | 2.09 | 11.6 [A] | 2,52 |
| 2019 | 12.2 [A] | 2.07 | 10.1 [A] | 1.57 | 20.3 [B] | 1.23 | 11.1 [A] | 1.79 |
| Average | 15.1 | | 12.0 | | 16.4 | | 11.1 | |
| **Moisture content in harvested biomass (%)** | | | | | | | |
| | **Lukavec** | | | | **Průhonice-Michovky** | | |
| | **M1** | | **M6** | | **M1** | | **M6** | |
| **Year** | **Autumn** | **Spring** | **Autumn** | **Spring** | **Autumn** | **Spring** | **Autumn** | **Spring** |
| 2008 | 38 | | 35 | | 53 | | 59 | |
| 2009 | 51 | 11 | 54 | 10 | 50 | 28 | 44 | 13 |
| 2010 | 41 | 9 | 34 | 10 | 56 | 20 | 62 | 17 |
| 2011 | 49 | 33 | 47 | 22 | 41 | 25 | 34 | 24 |
| 2012 | 43 | 8 | 42 | 5 | 37 | 17 | 51 | 20 |
| 2013 | 25 | 7 | 20 | 7 | 26 | 14 | 39 | 19 |
| 2014 | 45 | 11 | 41 | 10 | 44 | 21 | 35 | 31 |
| 2015 | 42 | 9 | 35 | 9 | 50 | 18 | 42 | 36 |
| 2016 | 57 | 8 | 42 | 6 | 35 | 21 | 31 | 12 |
| 2017 | 71 | 8 | 68 | 6 | 47 | 16 | 32 | 29 |
| 2018 | 32 | 8 | 37 | 6 | 50 | 22 | 51 | 15 |
| 2019 | 32 | 9 | 37 | 8 | 41 | 20 | 28 | 29 |
| 2020 | | 9 | | 9 | | 10 | | 11 |
| Average * | 44 | 11 | 41 | 9 | 44 | 19 | 42 | 21 |

Table legend: s—standard deviation; [†] years are considered as financial years (I-XII); * average yields from 12 harvests are calculated for 13 years of the experiment; [A–B] indexed capital letters show results of statistical analysis of yields (MANOVA–Duncan; homogenous groups) in individual years and sites.

### 3.5. Invasive Behavior

Observations of experimental sites in Průhonice (Michovky, Zelinářská zahrada) resulted in important findings confirming the ability of selected *Miscanthus* clones to spread from the original planting site spontaneously (Figure 2a). We have found relatively large volumes of tens to hundreds of well-growing seedlings of *Miscanthus sinensis* and/or *Miscanthus* ssp. with one individual found ca. 60 m from the experimental site on permanent grassland. By the end of the growing season, around 200 new *Miscanthus* seedlings were found in surrounding growths up to 80 m away. These seedlings were confirmed only on cultivated sites with bare soil. These rooted seedlings were one or two years old, from 5 to 20 cm tall with 1–4 stems. In 2009, samples of 35 individuals of rooted seedlings were taken to find their parental plants. All *Miscanthus sinensis* and *Miscanthus* ssp. clones selected for experimental germination verification could germinate, although in varying volumes depending on the clone. This ability appears more or less risky from the perspective of standard sexual reproduction and potential spreading in neighborhoods. Clones classified as *Miscanthus* × *giganteus* and *Miscanthus sacchariflorus* were sterile in our climatic conditions. For *Miscanthus sacchariflorus*, sterility can be caused locally by the absence of suitable pollinating plants in the time of flowering or unfavorable temperatures—the tested clone flowers very late. *Miscanthus sacchariflorus* had intensive vegetative reproduction via rhizomes. During our experiment, rhizomes spread 1.5–2 m within a season, presenting a risk in terms of nature protection and field management.

Based on the evaluated experimental results, i.e., the combination of triploidy and verification of the seeds' inability to germinate—the least risky clones, if we consider invasive behavior, are clones M1, M2 (*Miscanthus* × *giganteus*), and also, the triploid genotype M9 (*Miscanthus sinensis* 'Goliath'). However, when seed germination tests were repeated, the M9 clone showed a slight but certain ability to germinate (similar results were published by [53]).

### 3.6. Economic Analysis

Results of the calculation of the minimum price with and without subsidies (SAPS-Single Area Payment Scheme) are shown in Table 7. The minimum price of *Miscanthus* biomass was calculated to be between 3.0–15.2 EUR/GJ for the subsidized variant and 4.3 to 23.1 EUR/GJ for the variant without subsidies, in both cases, depending on the yield curve. The project's cost structure for the entire lifespan of the *Miscanthus* stand (in current values) is shown in Figure 5.

**Table 7.** Results of minimum price modeling of biomass of *Miscanthus* × *giganteus* for six yield curves (Yc 1–6) with and without subsidies (SAPS).

| Yield Curve | Average Yield * | Minimum Price with SAPS ** | Minimum Price without SAPS ** |
|---|---|---|---|
| | t DM ha$^{-1}$ y$^{-1}$ | EUR/GJ | EUR/GJ |
| Yc 6 | 12 | 3.0 | 4.3 |
| Yc 5 | 10 | 3.4 | 5.0 |
| Yc 4 | 8 | 4.2 | 6.2 |
| Yc 3 | 6 | 5.4 | 8.0 |
| Yc 2 | 4 | 7.8 | 11.8 |
| Yc 1 | 2 | 15.2 | 23.1 |

Table legend: * average yield is the average of expected yields during 10 (winter) harvests after establishment; ** the economic model was calculated in Czech Crowns (CZK), and the results recalculated to EUROs using the exchange rate of 1 EUR = 25 CZK.

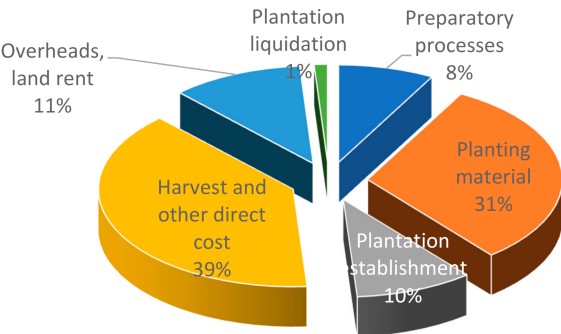

**Figure 5.** The cost structure of the *Miscanthus* biomass production divided into categories (in current values, with a nominal discount of 10%, Yc 5 = 10 t DM ha$^{-1}$ y$^{-1}$).

Thus, the minimum price is the price of (raw) biomass after removal from the field to the storage location (distance up to 10 km). Additional costs related to the storage of biomass and delivery to the final consumer (e.g., to the heating plant) or its supply for reprocessing into solid biofuels (pellets, briquettes) are not included. The minimum price also reflects only the losses of biomass in harvesting and transport to the place of storage, but not from storage, or subsequent transport or eventually, processing. Information on the impact of biomass losses and storage of biomass on the minimum price can be found, e.g., in [73].

The costs of establishing a plantation, including planting material, are high and significantly affect the minimum price. Figure 5 shows that this represents 44% of all project expenditure (in present value). Some reduction in the minimum price can be achieved by extending the life of the plantation, which then spreads the cost of establishing the plantation over a longer period. This can be documented with the variant of the calculation, where the lifetime of the stand is assumed to be 15 years and the continuation of a gradual decrease in yields (by about 1/3 in the 15th year compared to 10 years). The minimum biomass price for the Yc-4 yield curve decreases from 4.2 EUR/GJ to 3.9 EUR/GJ. Similarly, in the Yc-5 yield curve, there is a slight decrease from 3.4 EUR/GJ to 3.2 EUR/GJ (both calculations assume SAPS subsidy). This small reduction in the minimum price is mainly because the high costs of establishing a stand are spent at the beginning of the plantation project, while the maximum yields and thus, sales, are achieved only over time. Similarly, extending the life of a plantation generates additional cash flows, but the value of these contributions has little weight on the project's overall effectiveness. The key aspect of improving the economic effectiveness of *Miscanthus* is to reduce the cost of establishing plantations.

The above mentioned process used to determine the minimum price of biomass presents only one of three possible ways to look at the price of biomass.

The second perspective would be to look at it from the producer's point of view (supply side), where, in practice, the producer growing biomass for energy purposes would accept only a price for this biomass that would ensure, at least, a similar economic effect as growing conventional crops. Due to the high subsidies for growing conventional crops and the relatively high prices given for these conventional crops, the price, then, that producers would require for intentionally grown biomass is thus significantly higher. A detailed discussion of this aspect can be found, e.g., in [74]. According to the latest calculations carried out by the authors (price level 2019), high revenues from conventional crops lead to a biomass price increase from the minimum price for biomass from *Miscanthus* crops in comparison to the values given in Table 7, approximately in the range from 12–70%, with an average value of about 42% for all areas.

The third point of view on the price of biomass is that of the buyer (demand). Here, the price that the purchaser is willing to pay for intentionally planted biomass from energy crops does not exceed the price of other fuels (e.g., biomass from other sources such as forest biomass). In many cases, using (burning) raw biomass directly in bale form is

not possible and can only be expected for larger heating plants or those equipped with the appropriate technology. For smaller or local heating plants, such biomass must be transformed into solid biofuels—pellets or briquettes. However, manufacturing pellets and briquettes significantly increases the price of biomass, e.g., the cost of pelletization in the Czech Republic is about 4.5–5.0 EUR/GJ, which further significantly increases the (minimum) price of biomass [73].

The limit price of raw biomass from *Miscanthus* plantations in Czech conditions is estimated to range from 6–8 EUR/GJ, assuming price acceptability from the demand side. This is especially valid if the biomass is further processed into solid biofuel, and therefore, the price has been increased due to pelletization. The limit price of biomass in pellet form is, here, the price of wood pellets, i.e., 11.2 EUR/GJ minus the cost of pelletization—for more information, see [75]. If we consider the direct combustion of biomass (straw bales) in large power plants or cogeneration plants, the limit of the biomass price here will be influenced by the amount of support for electricity from direct combustion of biomass that varies according to the year of commissioning and category, see [76]. From the current amount of electricity support in the form of FIP (feed-in-premium) tariffs for intentionally grown biomass combustion, it is possible to estimate a biomass limit price of 6–7 EUR/GJ.

## 4. Discussion

After twelve consecutive harvest years, results on our two sites show yields similar to those in other experimental plantations in Central-Eastern European conditions and low-input agronomy [9,39,77,78]. For instance, [39] observed a mean yield of dry biomass of 13.7 t DM ha$^{-1}$ y$^{-1}$ in a non-fertilized variant of an 11-year experiment with *Miscanthus $\times$ giganteus* in southern Germany. The mean biomass yield of M1 and M2 clones (*Miscanthus $\times$ giganteus*) during spring harvests in Lukavec was 10.8 resp. 10.4 t DM ha$^{-1}$ y$^{-1}$. In the Průhonice-Michovky site, it was 13.5 respectively 14.7 t DM ha$^{-1}$ y$^{-1}$. These yields are comparable, if not slightly better than other new lignocellulose energy crops like poplar, willow, reed canary grass, or Schavnat in Czech conditions [79–81].

Even though spring harvests have lower yields than in autumn, they can be recommended because the concentrations of potassium, chlorine, nitrogen, and sulfur in *Miscanthus* biomass decreases significantly due to the translocation of nutrients to the root part and its leaching during winter [82]; a similar result was also recently confirmed by [83]. In comparison with woody energy crops (poplar, willow), *Miscanthus*' spring harvest biomass is less suitable for direct burning in some, especially smaller boilers, where it can create slagging in the heat exchanger. The effectiveness of *Miscanthus* biomass can be improved by mixing it with woodchips to produce pellets [84].

*Miscanthus*, however, can also be important as an effective source of commodities and materials, e.g., chipboard, pellets for animal (pet) bedding, cement particle boards, biocomposite automotive component, or biogas production from autumn (green) harvest, that have higher added value than energy biomass [85–87].

Knowledge about the invasiveness of some genotypes, resp. species of *Miscanthus* in European conditions, have been taken into account [88] by breeders, and it can be expected that new varieties will be 'minimum or zero invasive' for both generative and vegetative ways of reproduction and dispersal into the surrounding fields and countryside. In the Czech Republic, only clones of *Miscanthus $\times$ giganteus,* a non-invasive triploid, can be used in agriculture practice [56]. Since 2010, all clones of *Miscanthus sinensis* have been excluded from the "List of plants suitable for cultivation of energy biomass from the point of view of minimizing risks to nature and landscape protection" [89], which is a methodological support tool for decision making in nature protection regarding the use of non-native energy crops in the landscape.

At present, there are economic barriers to the faster development of *Miscanthus* cultivation. Competition with conventional (annual) crops is the main barrier that has the following aspects:

(1)  In contrast to conventional crops, *Miscanthus* plantations have high one-off costs for stand establishment. These one-off costs represent around 1/3 of the total cost for the *Miscanthus* stand (in present value) over its entire life cycle (10 years). In this way, the grower must, at the outset, invest significantly more money per unit of area than in the case of conventional agricultural production.

(2)  The maximum production of biomass is reached up to 2–3 years after establishment, which, from the producer's point of view, means that cash flow is initially worse.

(3)  Having multiyear plantations of energy crops is significantly riskier for producers, both in terms of the higher one-off costs of establishing the stands and losses after establishment due to crop damage or possible changes in the biomass market. An investor or farmer of perennial energy crops cannot react as quickly to market changes as someone who has invested in conventional crops with a one-year production cycle. One reason for this is that most agricultural land in the Czech Republic is still farmed on leased land (about 70%—see [90]), and rental periods are generally shorter than the life cycle of the energy crop plantation, thus further increasing the risk.

Another significant economic barrier is the relatively high costs related to growing biomass in a *Miscanthus* plantation. The minimum price of produced biomass (with a 10% nominal discount) assumed using average to less fertile soils ranges from 5.4–7.8 EUR/GJ, i.e., 57–73 EUR $t^{-1}$ of fresh matter [87] have calculated prices of 35–47 EUR $t^{-1}$ for *Miscanthus* biomass for direct combustion in German conditions, but for higher yields (15–25 t DM $y^{-1}$), intensive agronomy (fertilization, density), and much longer plantation lifetime (20 years). Farmers, however, in practice, would demand an even higher price that would at least give them the same economic effect that they would have from growing conventional crops, thus increasing the price of raw biomass by, approximately, an additional 43% (on average).

The price of raw *Miscanthus* biomass (without transport, storage, or processing costs (into pellets or briquettes) significantly exceeds the limit of the competitive price of raw biomass estimated at 6 (max. 8) EUR/GJ in the Czech Republic. This limit is important, as can be seen from the results of the authors' analyses, which show that because the minimum price of biomass increases due to the competition from conventional energy crops, there is no land on which any farmer would want to establish *Miscanthus* stands and accept 6 EUR/GJ or less. If the limit price would be 8 EUR/GJ, then producers would consider establishing *Miscanthus* stands on approximately 27% of the Czech Republic's agricultural land.

Another barrier is on the consumers' side and their technological limitations. To date, it has been technologically easier for consumers to focus on woody biomass rather than straw biomass that would need further investment into a suitable boiler using straw fuel. Otherwise, straw biomass would need to be made into pellets or briquettes, which significantly increases the price of the produced biofuel.

Economic barriers to the development of *Miscanthus* plantations (or other perennial crops) can be reduced by the following:

- Providing targeted subsidies for plantation establishment to decrease the investor's risk.
- Supporting long-term contracts to purchase biomass for energy crops using a price formula.
- Using plantations of perennial energy crops for additional benefits, i.e., non-production functions (e.g., decreasing soil erosion, phytoremediation, increasing the soil's humus content and water capacity).

Another measure that would significantly increase, albeit indirectly, the competitive ability of intentionally grown biomass against conventional fuels is to increase markedly the carbon costs (e.g., in the form of an emission allowance or carbon tax) included in the price of fossil fuels.

## 5. Conclusions

Average yields of *Miscanthus × giganteus* clones tested in our experiment (M1, M2 ≥ 10–15 t DM $ha^{-1}$ $y^{-1}$) are comparable, if not slightly better than other new

lignocellulose energy crops (poplar, willow, or Schavnat) in Czech conditions. *Miscanthus × giganteus* clones have good potential for commercial production of energy biomass, especially in warmer regions of Central and Eastern Europe (average annual daily temperature °t ≥ 9–10 °C) with an annual sum of precipitation above 500–550 mm.

Results of monitoring *Miscanthus × giganteus* yields and the course of weather during our experiment (13 years) have shown that *Miscanthus × giganteus* adapts well to dry years (or its parts) characterized by low precipitation ($\sum$P = 350–450 mm y$^{-1}$) and increasing annual daily temperatures (average annual daily temperature °t ≥ 10.5 °C).

Clones of *Miscanthus sinensis* tested in our experiment could not be recommended for energy biomass production due to their strong invasive ability. The sterile triploid clones of *Miscanthus × giganteus,* however, have been recommended with minimum risks for nature and landscape. Some clones of *M. sinensis* have shown the potential to be bred for colder conditions.

Results of economic modeling have shown that there are significant economic barriers to the development of perennial energy crops, especially those resulting in straw biomass. These barriers not only include the current and relatively high profitability of conventional annual crops, which in turn increases the expected price of biomass from energy crops, but also the economic risk associated with the large portion of one-off initial establishment costs. The competitive ability of straw biomass is significantly lower because of the consumers' technological limitations that usually do not enable them to burn straw biomass directly. Burning straw biomass then is taken into consideration only by larger heating or cogeneration plants. Smaller or local plants need biomass in pellet or briquette form, which means an increase in price and a decrease in competitive ability. At these smaller plants, biomass (processed into pellets or briquettes) can be competitive if natural gas is not available or where using a heat pump instead of a coal furnace is not relevant due to the high costs of reconstructing the heating system.

Regarding the article's question, "Can *Miscanthus* fulfill its potential as a new biomass crop—for energy and material in the Czech Republic (and CEE countries)?", our team would answer positively, but only if the following conditions and steps would materialize in the upcoming years:

- Improvement of *Miscanthus × giganteus* gene pool (new varieties) and agrotechnology (to lower establishment cost, prolong production period to 15–20 years, improve the precision of fertilization, minimize the invasive risk) continues.
- Climate change trends continue with growing effects of weather extremes and changes (droughts, temperature growth) in CEE countries, which may improve growing conditions for *Miscanthus* (C4 plant) over conventional crops (mostly C3 plants).
- A new approach of EC or member states to current agriculture subsidy policy (CAP), which would evaluate environmental services of *Miscanthus* and other new biomass crops, is implemented.
- Further development of the bioeconomy in the EU occurs, thus increasing demand for *Miscanthus* biomass for utilization in products with higher additional value, e.g., construction materials, industrial products, and second-generation biofuels.

**Author Contributions:** Conceptualization, J.W., J.B., K.V. and J.K.; methodology, J.W., Z.S. (field research) and J.K. (economy); validation, J.W., J.K. and K.V.; formal analysis, J.B.; investigation, J.W., J.B., K.V. and Z.S.; resources, J.B., K.V. and Z.S.; data curation, J.W., J.B. and K.V.; writing—original draft preparation, J.B., J.W. (field research) and K.V. (economic analysis); writing—review and editing, J.W. and J.K.; visualization, J.W.; supervision, J.W., J.K., K.V.; project administration, K.V.; funding acquisition, K.V. All authors have read and agreed to the published version of the manuscript.

**Funding:** This research was performed as part of project no. TD03000039 supported by TACR—the Technology Agency of the Czech Republic.

**Institutional Review Board Statement:** Not applicable.

**Informed Consent Statement:** Not applicable.

**Data Availability Statement:** The data and genotypes presented in this study are available on request from the corresponding author, Jan Weger, weger@vukoz.cz.

**Acknowledgments:** We would like to acknowledge Václav Veleta from the field research station in Lukavec (experimental site) for his long-term cooperation and provision of field data from the experimental *Miscanthus* plantation and Michal Severa for carrying out germination tests on *Miscanthus* seeds. Jana Jobbiková, Tereza Humešová, and Aleš Tobyška provided technical and administrative support of the field research; Janice Forry provided proofreading, translations, and language finalization of the text. We would also like to thank Iris Lewandowski, Uffe Jørgensen, and Jens Bonderup Kjeldsen for providing planting material of the original *Miscanthus* genotypes.

**Conflicts of Interest:** The authors declare no conflict of interest.

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
