# Peer review of "Can Miscanthus Fulfill Its Expectations as an Energy Biomass Source in the Current Conditions of the Czech Republic?—Potentials and Barriers"

_agriculture, doi:10.3390/agriculture11010040_

Round 1

Reviewer 1 Report

48-51 I agree with the authors that when it comes to rape and cereals, this ratio may be beneficial for non-food crops. But now the main first generation source is corn. Information is missing. How energy crops can compete with corn? Perhaps this relationship will be similar to that of rape, but ... rape is not the main first generation crops. 

50-51 incorrect citation records, [7], [8], [9], [69]

68- “(Stolarski, pers, comm.)” is it citation?

75-76 “schavnat and triticale (Picture 1)” Do I mean miscanthus and schavnat (Fig. 1)? If yes, please correct it.

82-92-  This information should be confirmed in literature reports.

105-107- These data also require literature confirmation. It would be best if it were reports from current research: Suggestions: Potential of bioethanol production from biomass of various Miscanthus genotypes cultivated in three-year plantations in west-central Poland. Cerazy-Waliszewska et al. 

126-139 Miscanthus sinensis, miscanthus  gigantheus… ITALICS! 

MATERIALS AND METHODS

172 Diagram 1? There should be “Figure 2”

206-207 When plant height was measured?

253- Picture 3??? Where is it? 

337- Picture 3??? Figure 3!

353- Table 5 no 6

420- Table 7 or 6?

Author Response

Response to Reviewer 1

Thank you very much for the comments and feedback on our article. We have accepted your comments and suggestions - see our responses (in Italisc) below your comments:

Response to Reviewer 1

Thank you very much for the comments and feedback on our article. We have accepted your comments and suggestions - see our responses below your comments:

48-51 I agree with the authors that when it comes to rape and cereals, this ratio may be beneficial for non-food crops. But now the main first generation source is corn. Information is missing. How energy crops can compete with corn? Perhaps this relationship will be similar to that of rape, but ... rape is not the main first generation crops. 

Response of authors: Thank you for the comment. We have not included corn in our article because its residual biomass (corn stalks, corn stover) is not very suitable for direct burning due to the high content of elements and impurities (soil dust, green matter). These tend to cause slagging in boilers, and economically, corn stover for direct burning is much less favorable than wheat and rape residual straw.

Maize is used in biogas plants, which have substantially decreased in number. Miscanthus, however, is used for direct combustion or production of solid biofuels. The increase in the maize planting area (for biogas) in the last 10-15 years has led to significant environmental problems associated with soil erosion and reduction of biodiversity. In the future, corn cultivation is expected to decline because of these environmental considerations.

50-51 incorrect citation records, [7], [8], [9], [69]

Response of authors: Citation records were corrected and transformed into the required form.

68- “(Stolarski, pers, comm.)” is it citation?

Response of authors: We have used information from personal communication with a national expert where there are no published data or analyses available on reasons for the decline of the Miscanthus growing area. We changed these to citations according to the article methodology.

75-76 “schavnat and triticale (Picture 1)” Do I mean miscanthus and schavnat (Fig. 1)? If yes, please correct it.

            Response of authors: Thank you for the comment. It was only meant to show the development of schavnat planting. We have moved the reference to  Picture  1 to  different place and added an explanation to avoid ambiguity.

82-92-  This information should be confirmed in literature reports.

            Response of authors: Official and scientifically reviewed information about the development of commercial plantations of second-generation energy crops is not available in the Czech Republic, except the planting area of SRC, which has its own category of agriculture land utilization in the LPIS. Therefore, data about other crops were collected via personal contact and communication by the main author.

105-107- These data also require literature confirmation. It would be best if it were reports from current research: Suggestions: Potential of bioethanol production from biomass of various Miscanthus genotypes cultivated in three-year plantations in west-central Poland. Cerazy-Waliszewska et al. 

Response of authors:  We have integrated the above-mentioned citation (Cerazy-Waliszewska et al., 2019) and (Grams et al., 2019) into the appropriate part of the article’s text.

126-139 Miscanthus sinensis, miscanthus  gigantheus… ITALICS! 

            Response of authors: Thank you for the comment. Scientific names were corrected in this paragraph.

MATERIALS AND METHODS

172 Diagram 1? There should be “Figure 2”

            Response of authors: Thank you for the comment. Designation was corrected accordingly.

206-207 When plant height was measured?

Response of authors: Heights were measured usually a few days or hours before autumn and spring harvests.“ We clarified the terms of height measurements.

253- Picture 3??? Where is it? 

Response of authors: Thank you for the comment. Designation was corrected

337- Picture 3??? Figure 3!

Response of authors: Thank you for the comment. Designation of pictures and figures was corrected

353- Table 5 no 6 JW –

Response of authors: Thank you for the comment. Designation was corrected

Reviewer 2 Report

Grasses of the genus Miscanthus are characterized by high biomass yields per unit area, which makes them highly valuable feedstock for bioenergy production. Therefore, the manuscript entitled “Can Miscanthus fulfill its expectations as energy biomass source in current conditions of Central Eastern European countries? - Potentials and barriers” addresses a topical issue, and the research findings have important scientific and practical implications. Such studies, involving an economic analysis and an evaluation of the invasive potential of grasses of the genus Miscanthus under the environmental conditions of Central and Eastern Europe, are scarce. Therefore, the present study is interesting and worth noting.

The manuscript is well organized and clearly written. The sections and subsections have a logical structure. The Introduction is appropriate and comprehensive, although subsection 1.2 should include a paragraph describing the use of giant miscanthus for biogas and bioethanol production. Several papers devoted to giant miscanthus biomass production for energy purposes in north-eastern Poland were published in 2016 – 2020, and they could be cited in this subsection as well as in the Discussion section. The objective of the study should also be clearly started in the Introduction section.

The study was based on the results of a long-term field experiment investigating Miscanthus giganteus and Miscanthus sinensis clones. The experiment is well designed and presented. The materials and methods are appropriate and sufficient to validate the research hypothesis. I only cannot agree with the statement (subsection 2.5, line 217) that the life-span of giant miscanthus is 8-14 years. According to the presented economic analysis, the life cycle of the analyzed crop species is only 10 years, i.e. very short. However, according to the scientific literature, giant miscanthus plantations can be used for a longer period of time (20-25 years), which is an important consideration in view of the high costs of plantation establishment and attempts to increase the competitiveness (profitability) of miscanthus biomass production, relative to other energy crops. The results of the study are well described, thoroughly analyzed, interpreted and compared with previously reported findings.

Author Response

Response to Reviewer 2

Thank you very much for the comments and feedback on our article. We have accepted your comments and suggestions - see our responses in Italics below your comments:

The manuscript is well organized and clearly written. The sections and subsections have a logical structure. The Introduction is appropriate and comprehensive, although subsection 1.2 should include a paragraph describing the use of giant miscanthus for biogas and bioethanol production. Several papers devoted to giant miscanthus biomass production for energy purposes in north-eastern Poland were published in 2016 – 2020, and they could be cited in this subsection as well as in the Discussion section.

Response of authors:  We have integrated the above-mentioned citation (Cerazy-Waliszewska et al., 2019) and also another (Grams et al., 2019) into an appropriate part of the article’s text.

The objective of the study should also be clearly stated in the Introduction section. 

Response of authors: Thank you for the comment. We have added a short chapter 1.4. the objective of the article.

The study was based on the results of a long-term field experiment investigating Miscanthus giganteus and Miscanthus sinensis clones. The experiment is well designed and presented. The materials and methods are appropriate and sufficient to validate the research hypothesis. I only cannot agree with the statement (subsection 2.5, line 217) that the life-span of giant miscanthus is 8-14 years. According to the presented economic analysis, the life cycle of the analyzed crop species is only 10 years, i.e. very short. However, according to the scientific literature, giant miscanthus plantations can be used for a longer period of time (20-25 years), which is an important consideration in view of the high costs of plantation establishment and attempts to increase the competitiveness (profitability) of miscanthus biomass production, relative to other energy crops. The results of the study are well described, thoroughly analyzed, interpreted and compared with previously reported findings.

Response of authors: Thank you for your opinion -  Based on your comment, we have calculated a variant of the economic model with a 15-year life-span of a Miscanthus plantation. We have included a new paragraph to comment on these results in chapter 3.5.  However, the impact of extending the life of the plantation had rather a limited effect on the minimum price (8-9%).

Reviewer 3 Report

Title - The title of the manuscript must be changed. The research was carried out only in the Czech Republic and not in the CEE countries.

Line 30 RES - the abbreviation should be explain

Line 50-51 should be [7-9]. Not found Stolarski 2019

Line 161 „Soil is modal Cambisol (brown soil)” – Authors should use the WRB classification 2015

Line 166 Table 1 I propose delete the information about „Valuation soil ecological unit”. This is clear only for readers from Czech Republic

Line 185 table 2 is ZelináÅ™ská zahrada and should be PrůhoniceZelináÅ™ská zahrada like in table 1Line Line 194-196 – Authors should explain why no fertilizers were used in Pruhonice? If rhizomes were planted on May, why wasn't nitrogen fertilization applied at two sites?

Line 212 is These samples were dried at 105°C until constant weight. Authors should explain why used so high temperature for plants? Usually we dry the plants at 60°C and the soil at 105°C.

Line 294 Table3 Authors should use full localization name like in table 1

Line 330 Table 4 No statistical estimation of the results for Průhonice 2016.

Line 340 Figure 3 - Figure 3 is not clear. How the yield curves 1 - 6 were calculated. The differences between yields are 7 times. Why?

Line 346-354 This information should be included in the table. Why the table 5 does not provide data on the yield and moisture content depending on the harvest date - these are the basic information affecting the suitability of this plant for energy use? Was the content of minerals in plants not dependent on the maple, year of research and harvest date, as described in the text above dry weight and humidity? Average should be removed from the table because it is too high averaging. Statistical elaboration of the results should be includ in table 5.

Line 355-356 Table 5. The content of elements in Miscanthus biomass is several times lower than that stated in the literature. The authors should explain why the content of elements in the plant is so low.

Line 548-561 The research was carried out for the conditions of the Czech Republic, not Central Eastern Europe. Based on the research results, no conclusions can be drawn about the cultivation of Miscanthus in CEE countries

Line 795 Authors should provide full citation

Author Response

Response to Reviewer 3

Thank you very much for the comments and feedback on our article. We have accepted your comments and suggestions - see our responses in Italics below your comments:

Line 30 RES - the abbreviation should be explain

Response of authors: abbreviation explained

Line 50-51 should be [7-9].

Response of authors: citations corrected

Line 161 „Soil is modal Cambisol (brown soil)” – Authors should use the WRB classification 2015

Response of authors: correct name Cambisol has been used without any redundant information or qualifiers according to WRB classification

Line 166 Table 1 I propose delete the information about „Valuation soil ecological unit”. This is clear only for readers from Czech Republic

Response of authors: We agree – the row with info about ‘Valuation soil ecological unit’ was deleted.

Line 185 table 2 is ZelináÅ™ská zahrada and should be PrůhoniceZelináÅ™ská zahrada like in table 1

Response of authors: Corrected accordingly

Line Line 194-196 – Authors should explain why no fertilizers were used in Pruhonice? If rhizomes were planted on May, why wasn't nitrogen fertilization applied at two sites?

Response of authors: Thank you for your comment. The concept of the experiment - based on experience with other perennial energy crops and also some literature - was to use the best available, but low-input agrotechnology for the following reasons - our sites had a good level of nutrients, and fertilization would support growth of weeds in the  first years. Not fertilizing would minimize the risk of nutrient leaching (N, P). We aimed to test production with minimized inputs and until now (14 harvests) have yet to experience symptoms of nutrient shortage. We are now considering the first application of fertilizers to utilize increased soil humus and  capacity for absorption of nutrients. We have included an  explanation of our concept in the article.- this paragraph was reformulated to explain our concept.

Line 212 is These samples were dried at 105°C until constant weight. Authors should explain why used so high temperature for plants? Usually we dry the plants at 60°C and the soil at 105°C.

Response of authors: We use maximum temperature of 105°C which is standard for drying energy biomass, because at this temperature fuel characteristics of biomass are not influenced (esp. volatile substances are not released (from 135 °C).  .- a sentence was added to explain the selection of temperature.

Line 294 Table3 Authors should use full localization name like in table 1

Response of authors:  Full names were added - where missing - of experimental sites. Localization Průhonice was left only at table of climatic conditions because they are the same for both sites (Průhonice-Michovky, Průhonice-ZelináÅ™ská zahrada) as they are 1.4 km away from each other.

Line 330 Table 4 No statistical estimation of the results for Průhonice 2016.

Response of authors:  results of statistical evaluation were added for the year 2016.

Line 340 Figure 3 - Figure 3 is not clear. How the yield curves 1 - 6 were calculated. The differences between yields are 7 times. Why?

Response of authors:  Thank you for your question and comment. As explained very briefly under Figure 3, yield curves (Yc 1-6) show expected yields of Miscanthus plantation in commercial practice on different growing zones. They - Yc and growing zones - were created by (energy) crop experts using field data (yields) from plantations established on different sites (climatic and soil conditions) each characterized by detailed „Valuation soil ecological unit” of the Czech agricultural land valuation system.  We have added a new paragraph and citations under Figure 3  which explain the methodology used to create these yield curves.  

Line 346-354 This information should be included in the table. Why the table 5 does not provide data on the yield and moisture content depending on the harvest date - these are the basic information affecting the suitability of this plant for energy use? Was the content of minerals in plants not dependent on the maple, year of research and harvest date, as described in the text above dry weight and humidity? Average should be removed from the table because it is too high averaging. Statistical elaboration of the results should be includ in table 5.

Response of authors:  see next response.

Line 355-356 Table 5. The content of elements in Miscanthus biomass is several times lower than that stated in the literature. The authors should explain why the content of elements in the plant is so low

Response of authors:  Thank you for your valuable comment. After we have corrected the units in Table 5, which were mistakenly given as g/kg  instead of %, nutrient and element contents are mostly in the range found in literature.  Nevertheless, we have decided to exclude the table, firstly, because this topic is not the main objective of the article, and secondly, because it was analyzed in detail in a recent article of co-author Z. Strašil, including data from the original table 5. Instead, we have added a table with biomass yields and moisture contents in autumn and spring harvests, which are more related to the objective of our article.

Line 548-561 The research was carried out for the conditions of the Czech Republic, not Central Eastern Europe. Based on the research results, no conclusions can be drawn about the cultivation of Miscanthus in CEE countries

Response of authors:  Thank you for your expected comment. After thorough discussion in our team we have decided to change the name of the article to “Can Miscanthus fulfill its ….. in current conditions of the Czech Republic - Potentials and barriers” Though our initial intent was to complete a biological and economic analyses to reach a conclusion regarding Miscanthus utilization in the wider region of CEE states, we came to the conclusion that especially the economic conditions and environmental goals of these individual states are now much more diverse and that we would need much more information regarding rational evaluation of Miscanthus potential in the CEE region.    

Line 795 Authors should provide full citation

Response of authors:  We have updated all citations according to the journal manual using citation SW Mendeley.

Reviewer 4 Report

The authors describe the merits and demerits of plantation of Miscanthus as an energy biomass crop in Central Eastern European countries such as Czech Republic.  Issues of a paper is very interesting.  Long term evaluation of biomass production in Miscanthus genotypes is very useful information for a research area of bioenergy biomass crops.  However, quality of a paper is poor.  It has not well written as a scientific original paper.  This paper is still like a technical report, not scientific paper.   Aim of the present study, discussion and originality (new findings) are not clear.  Also, logic of a paper is unclear.  Authors did not mention of utilization of Miscanthus as raw materials as well. 

Author Response

Response to Reviewer 4

Thank you for the comments and feedback on our article. See our responses in Italics below your comments:

The authors describe the merits and demerits of plantation of Miscanthus as an energy biomass crop in Central Eastern European countries such as Czech Republic.  Issues of a paper is very interesting.  Long term evaluation of biomass production in Miscanthus genotypes is very useful information for a research area of bioenergy biomass crops.

However, quality of a paper is poor.  It has not well written as a scientific original paper.  This paper is still like a technical report, not scientific paper.   Aim of the present study, discussion and originality (new findings) are not clear.  Also, logic of a paper is unclear.  Authors did not mention of utilization of Miscanthus as raw materials as well. 

Response of authors:  The article was substantially upgraded. We have added new paragraphs, chapters and tables to support/describe in more detail our main objectives, use of original methods and approaches e.g. long-term monitoring of Miscanthus production under low-input agronomy; evaluation of invasive risk; original economic model for evaluation of Miscanthus competitiveness. We have removed some texts and one table with redundant information that do not meet the objectives of our article. We believe that we have identified the  main existing barriers (cost of biomass, invasive risk) for Miscanthus to become more attractive as an agriculture crop in the Czech Republic and other CEE countries. We have also documented in our field experiment the very good adaptability of selected Miscanthus clones to changing climate conditions (droughts, increased temperatures) in the region of CEE.

We have stressed the potential of material utilizations of Miscanthus biomass in the Introduction chapter (also with new citations), Discussion (pellets for animal (pet) bedding, cement particle boards, biocomposite automotive components) and in Conclusions “products with higher additional value, e.g. construction materials, industrial products…”

Round 2

Reviewer 3 Report

After proofreading, the manuscript can be published in MDPI Agriculture

Author Response

Response of authors: The manuscript was thoroughly proofread again by Mrs. Janice Forry  a native speaker, professional translator, and experienced editor of scientific manuscripts.  She has worked for several publications, including for several years as a language editor for the scientific journal Folia Geobotanica.

We have also added 3 more people to the acknowledgements.